# Determining Wheel Forces and Moments on Aircraft Landing Gear with a Dynamometer Sensor [note 1]

**DOI:** 10.3390/s20010227

**Published:** 2019-12-31

**Authors:** Jaroslaw Pytka, Piotr Budzyński, Tomasz Łyszczyk, Jerzy Józwik, Joanna Michałowska, Arkadiusz Tofil, Dariusz Błażejczak, Jan Laskowski

**Affiliations:** 1Faculty of Mechanical Engineering, Lublin University of Technology, Nadbystrzycka 36, 20-618 Lublin, Poland; j.pytka@pollub.pl (J.P.); tomasz.lyszczyk@gmail.com (T.Ł.); j.jozwik@pollub.pl (J.J.); 2The State School of Higher Education, The Institute of Technical Sciences and Aviation, 22-100 Chełm, Poland; jmichalowska@pwsz.chelm.pl (J.M.); atofil@pwsz.chelm.pl (A.T.); 3Department of Construction and Usage of Technical Devices, West Pomeranian University of Technology in Szczecin, 70-310 Szczecin, Poland; Dariusz.Blazejczak@zut.edu.pl; 4Faculty of Management, Lublin University of Technology, Nadbystrzycka 36, 20-618 Lublin, Poland; jlasko@wp.pl

**Keywords:** airfield performance, landing gear, wheel, force and moment measurement, strain gage, wheel force sensor, wireless data transfer, grassy airfield, GARFIELD System

## Abstract

This paper describes airfield measurement of forces and moments that act on a landing gear wheel. For the measurement, a wheel force sensor was used. The sensor was designed and built based on strain gage technology and was embedded in the left landing gear wheel of a test aircraft. The sensor is capable of measuring simultaneously three perpendicular forces and three moments and sends data to a handheld device wirelessly. For the airfield tests, the sensor was installed on a PZL 104 Wilga 35A multipurpose aircraft. The aircraft was towed at a “marching man” speed and the measurements were performed at three driving modes: Free rolling, braking, and turning. The paper contains results obtained in the field measurements performed on a grassy runway of the Rzeszów Jasionka Aerodrome, Poland. Rolling resistance of aircraft tire, braking friction, as well as aligning moment were analyzed and discussed with respect to surface conditions.

## 1. Introduction

Airfield performance of an airplane plays an important role in the analysis of takeoff and landing and determines, among others, ground roll distances. Forces and moments that act on aircraft landing gear wheels are effects of gravitational acceleration, as well as surface reactions. These reactions are easily obtainable on paved runways, but difficulties arise when an airplane operates on unpaved, grassy, or gravel surfaces. One method is to monitor surface mechanical data, related to weather conditions. The *GARFIELD* information system [1] is based on a wheel–grass model that includes analysis of wheel–soil (wheel–grassy surface) interactions considering soil modeling with a special caution to loads by aircraft tires [2]. The model will respect non-linear, dynamic effects such as hyper-elastic tire deflection, rheological soil response to high rate deformation, and effects of grass and roots. The wheel–grass model also employs the analysis of weather impact upon mechanical characterization of runway surface, which can be identified and verified by means of wheel–force measurements in full-scale tests [3]. 

Researchers dealing with flight tests of aircraft, piloted or unmanned, pay significant attention to the importance of ground tests, including measurements for the validation and verification of landing gear systems [4,5,6].

Significant emphasis is placed on the development and research of new measuring technologies and, as an example, a new solution in the field of strain gauge sensors can be cited. Pecora et al. have presented an innovative strain gauge for monitoring of inflatable structures, suitable for an use in aerospace [7]. Another interesting example is given by Petritoli et al. (2019-1), who investigated uncertainty and measured errors of an inertial navigation systems for UAV (Unmanned Aerial Vehicle) [8]. Petritoli et al. (2018) and Petritoli et al. (2019-2) targeted the problem of collocation of the sensors to be installed in a UAV with a reduced space, together with the allocation of the payload [9,10]. This problem is vital for full size aircrafts, not only typically smaller UAVs, especially if a sensor has to be embedded in an existing part of an aircraft. Eling et al. (2015) presented an innovative georeferencing system for the guidance, navigation, and control of UAVs [11]. This is an example of an inertial measurement unit (IMU)-based system, the method of a wide spectrum of applications in aerospace.

Measurement of forces and moments that act on a road wheel is a well known practice in automotive research and development. It is usually done either on laboratory test stands or on a test vehicle. Either way, a device that provides wheel forces and moments is a wheel dynamometer. A wheel dynamometer is a measuring device that enables to measure forces and moments acting on a wheel in real time on a moving vehicle. The majority of wheel dynamometer solutions comes from the automotive industry or research, and a typical solution has a form of a 6-element sensor that measures three orthogonal forces, *F_X_*, *F_Y_,* and *F_Z_,* together with corresponding moments, *M_X_, M_Y_,* and *M_Z_* [12,13,14,15]. The measurement principle is based either on the strain gage or piezoelectric sensor technology. Some selected works that describe the use of such dynamometer systems for measurement of wheel forces and moments in various applications are included in the reference list [16,17,18,19].

The horizontal force, *F_X_*, measured by a wheel dynamometer can be correlated with bending or shearing the grass plants, as well as soil deformation. The braking friction force can be correlated with sliding over the grass. The vertical force, *Fz*, depends the grass compression and can be used for coefficient calculations (braking friction, *μ*, and rolling resistance, *k_RR_*, coefficients, as well as aligning moment, *M_Z_*). This approach is alternative to commonly used laboratory drop tests of landing gear [20,21,22]. The aim of this study was to apply a wheel force and moment sensor for measurement of aircraft landing gear on grassy surface. The primary paper, entitled *Measurement of Forces and Moments Acting on Aircraft Landing Gear Wheel*, was presented during the 2019 IEEE International Workshop Metrology for Aerospace in Torino, Italy [23].

## 2. Materials and Methods

### 2.1. Wheel Dynamometer System

The wheel dynamometer system used in the experiment consisted of a sensor, modified wheel and tire, wireless data transfer unit, and software. The sensor core was made of steel, and strain gages were used as sensing elements. The instrumentation amplifiers, which prepare the outgoing signals for data acquisition, were placed outside the sensor in a small box, together with a board of the microprocessor data analysis and transfer system. Figure 1 shows a schematic of the sensor, but without the electronics. This schematic also includes the orientation of forces and moments.

As mentioned earlier, the sensor was designed based on the strain gage measurement technique. The sensing element was made of spring steel and linear type strain gages were glued onto it.

The in-wheel transducer measures two forces, *F_Z_* and *F_X_,* and two moments, *M_Z_* and *M_X_*. Due to the restricted space mentioned above, a braking moment sensor was made as a separate, external device. The *M_Y_* moment was measured with the strain gages sealed on the brake jig, which is bending proportionally to braking moment. Figure 2 shows the complete sensor system, embedded in an aircraft wheel.

The wireless data transfer unit was built as a microprocessor system (see Figure 3), with an 8-channel A/D converter and a Bluetooth radio device to transfer measured data wirelessly.

The complete and detailed description of the wheel force sensor system is given in the reference by Pytka et al. [3].

### 2.2. Airplane Used in the Experiment

The PZL 104 Wilga 35 multipurpose airplane was used as a test aircraft. This is a light, four-place, multipurpose aircraft, powered by a 192 kW radial piston engine. The Wilga has a good reputation as a STOL (Short Takeoff and Landing) or even the so-called “bush-plane”, thanks to its landing gear. The aircraft has a tail-dragger-type landing gear and the main gears are rocker type with oleo-pneumatic shock absorbers. The landing gear wheels have low-pressure 500 × 200 mm tires with hydraulic brakes. The wheels are castored and have a positive rake angle of 18°; the axle offset is 400 mm. The airplane is 8.10 m in overall length, with a wingspan of 11.12 m, a wing area of 15.5 m^2^, and an empty mass, equipped, of 900 kg. In the landing tests, there were four persons on board and the take-off weight was 1150 kg. The test airplane is depicted in Figure 4. 

The wheel force sensor was installed on the test aircraft before measurements. Installation did not require any modification to the airframe or landing gear, since the sensor system was developed specially for the given aircraft. Simply, leveling up the aircraft, removing the wheel with its axle, and placing the sensor. Also, the brake hydraulic system had to be re-switched to the sensor, since the *F_Y_* wheel moment was measured by means of a transducer embedded in the wheel brake. 

Another element of the measuring system was the electronic unit, which consisted of five differential signal amplifiers, each for one of the five channels (two forces and three moments), as well as of a microcontroller-based, wireless data transfer unit. The electronic box was mounted on the landing gear with glue tape and the sensor was connected with the electronics with signal cables. The whole system, the wheel sensor, and the electronics power supply (7.2V 800 mAh NiMH accumulator) was placed in the cockpit. Figure 5 and Figure 6 show the details of the test airplane with the installed measuring equipment.

### 2.3. Grassy Airfield Conditions

Measurements of wheel forces and moments were performed on a grassy surface of the Rzeszów Jasionka aerodrome, located in southeast Poland. The measurements were carried out in the area adjacent to the runway, with the surface and soil conditions identical to those on the grassy runway. The height of the grass was approximately 10 cm each time of the measurements.

The days in the autumn and spring months were selected, when there is a high probability of increasing humidity and lowering the traction properties of the grassland due to the influence of weather conditions (rainfall, low temperature, low sunshine). They were on 17 October, 5 November, 5 December, and 13 March. Table 1 gathers data describing atmospheric conditions prevailing on the days of measurement.

Before the test runs, we also measured soil moisture using a handheld TDR moisture meter. TDR is an acronym for time-domain reflectometry, and this is a technique that utilizes the phenomenon of different electrical permeability of porous, three-phase media, such as soil. The TDR meter is very easy to use. Measurement requires simply inserting test electrodes into soil and the instrument quickly gives results, which are collected in Table 1. Figure 7 shows the test engineer performing soil moisture measurement on the grass field. See also the reference by Pytka et al. for more details on the use the TDR sensor in terramechanics studies [24].

### 2.4. Field Procedures

Since the sensor system has not been certified for flight tests yet, we could only manage ground tests. In this study, it was aimed to determine wheel forces and moments acting on the aircraft’s wheel at low speed of taxiing over the grassy runway. We used an all-terrain vehicle (ATV) as a tractor vehicle and the aircraft was connected to this vehicle by means of cables. As mentioned earlier, four adult persons were in the test aircraft during all tests. An approximate speed of taxiing was 5 km/h. The aircraft performed the following maneuvers:Accelerating from stop to the taxi speed;Taxiing with a constant speed (about 20–30 m);Turning left or right;Braking to a full stop or to block the wheels.

We performed a minimum of five repetitions of all the maneuvers for every measurement time. Figure 8 shows a sample test run on the grassy surface.

The wireless data transfer system enabled that the test engineer could be anywhere, within a 100 m range of the airplane. Since the measurements were performed with all four passengers on board, the test engineer was in the airplane during measurements. We tried to collect data on a laptop computer, but it was more comfortable to gather the results on a handheld smartphone.

## 3. Results

Results obtained in the field experiments include courses of wheel forces and moments acting on the left wheel of the Wilga airplane, and we collected data from four days of measurements: 17 October, 5 November, and 5 December 2018, as well as 13 March 2019. The following is a presentation and analysis of the results obtained in the autumn–winter part of the measurements campaign (a so-called “dead season” in sport aviation) that was performed as a full-season measurement for the verification of the wheel–turf interaction model of the *GARFIELD* information system [1,2,23,24].

### 3.1. A General Presentation of the Results

Figure 9 shows time courses of the vertical *F_Z_* and horizontal *F_X_* forces that act on the landing gear wheel. The *F_Z_* shows almost constant values with low fluctuations, probably caused by vertical vibrations of the entire aircraft rolling over the grassy surface, and those values are close to the aircraft weight component on the wheel. The horizontal force *F_X_* exhibits negative values since the sensor has a defined orientation and a force of the direction backwards is indicated as negative. Moreover, the course of the horizontal force *F_X_* reveals effect of wheel function modes: Free rolling (the first second of the course), acceleration increasing of the force’s values (between 1 and approximately 4 s of the course), braking to a full stop (5–9 s), and, again, acceleration (from approximately 9 s of the course). The peak maximum value of the horizontal force *F_X_* reaches 2.7 kN, but the average value is much lower. 

Another sample result is shown in Figure 10, where we can see time courses of the three moments acting on the aircraft wheel. The character of these courses is very similar to those of Figure 9. These is a short period of free rolling, then acceleration, braking and stop, and finally acceleration again. The peak maximum values reach 150 Nm for the *M_Z_* moment, 280 Nm for the *M_X_*, and 250 for the *M_Y_* moment. The values of the moments acting on the landing gear wheel changes their sign at a moment of a change between two function modes, for example, accelerating–braking.

In the next sections, an analysis of the measured forces and moments is presented in order to determine coefficients that describe wheel–surface interactions, which are of importance for the analysis and modeling of airfield performance of an aircraft. 

### 3.2. Rolling Resistance Coefficient

Rolling resistance is a property of the tire–surface system and depends on many factors. In automotive technology, rolling resistance is determined for a tire rolling on a drum test stand and its coefficient takes essentially constant values, on the order of 0.010–0.012. In the case of a tire interacting with a deformable surface, such as grassy surface, rolling resistance is difficult to determine, mainly due to the changing conditions of the surface, which was mentioned in the introduction. In addition, the values of the rolling resistance coefficient of the tire–grass surface system can vary by as much as an order of magnitude, taking into account, for example, changes in soil moisture. The presence of vegetation (grass, roots), as well as the size and mass of green parts of vegetation, are also significant.

The rolling resistance coefficient, *k_RR_*, is defined as the ratio of horizontal force during free turning to vertical force [25]:*k_RR_* = *F_X_*/*F_Z_*(1)

Figure 11 shows an example of horizontal force, *F_X_*, which is input data for calculations (upper graph) and the rolling resistance coefficient, *k_RR_* (lower graph), value determined using Equation (1). Average values are marked on the graphs (*F_X_* = 1719 N and *k_RR_* = 0.275). When determining the *k_RR_* coefficient, it is important to correctly select a portion of the *F_X_* force waveform. Namely, it is a fragment where the waveform is determined (in Figure 11, up, this fragment begins after the first second of the wave).

The exemplary value of the *k_RR_* coefficient was determined to be 0.275, which in comparison to the aforementioned value for a car tire on a paved road is almost 25 times [25]. This is the effect of a soft surface and a low-pressure aviation tire. The impact of weather conditions on the determined values of the *k_RR_* coefficient will be presented in the next section.

### 3.3. Braking Friction Coefficient

The braking friction coefficient *μ* is another important factor. It determines the course of the braking process and the landing distance of the aircraft. In the case of the tire–paved system (asphalt, concrete), the friction coefficient usually reaches values between 0.6–0.9. For unpaved surfaces such as grass turf, the friction coefficient is usually lower and may be 0.1–0.5. Factors affecting the *μ* value are grass length and humidity: The longer the grass and the more water it contains, the lower the friction coefficient. By definition, the friction coefficient is determined on the basis of the following relationship:*μ* = *M_Y_*/*F_Z_* × *r_d_*(2)
where *r_d_* is the dynamic radius of the tire.

Figure 12 shows an example of the braking moment, *M_Y_*, course and the braking friction coefficient determined. As it can be seen, *μ* values oscillate around 0.22.

### 3.4. Aligning Moment

When driving on a curved track, the road wheel moves in such a way that the longitudinal axis of the wheel is deflected from the direction of the instantaneous speed vector by an angle, called the side slip angle. The reaction of a pneumatic tire to side slip is to generate a moment that tends to reduce this angle. This moment is called the stabilizing or aligning moment.

The sensor used in the tests directly measures the value of the aligning moment, it is the moment marked as *M_Z_*. Figure 13 shows the *M_Z_* course recorded during one of the turning maneuvers. The importance of the *M_Z_* moment is revealed during taxiing, especially at high speed. The higher the *M_Z_* value, the easier it is to maintain a straight direction, while too high a stabilizing moment can cause significant steering forces when cornering, especially when taxiing at low speed.

### 3.5. Effect of Weather Condidtions

As already mentioned, the study was carried out over four day—in October, November, December, and March. The purpose of such a research program was to check to what extent the worsening deterioration of weather conditions, in terms of lowering the external temperature and increasing humidity of air and soil, affects the measured values of forces and moments, as well as the resulting coefficients. Based on the measurements carried out, mean values of repetitions (minimum of five repetitions for each day) of individual measurements were determined. A collection of final data in the form of bar charts is presented in Figure 14, Figure 15 and Figure 16, where average values of rolling resistance coefficient, *k_RR_*, friction coefficient, *μ*, and stabilizing moment, *M_Z_*, are shown for four days, representative for particular weather periods.

The rolling resistance coefficient increases, and this is a significant increase. The explanation for that is quite obvious: In autumn, late autumn, winter, and early spring, there is a gradual increase in humidity of both air and soil, which causes increasing deformation of the surface and, as a result, an increase in rolling resistance. Although we did not construct any relationship between the rolling resistance coefficient and soil moisture (see Table 1), these parameters correspond well to each other.

The second term, braking friction coefficient shows an inverse relationship, i.e., it decreases during the period considered. In this case, the cause is similar, although the increasing humidity works slightly differently, namely, lower grip (=more slipperiness), especially in the case of grass-covered surfaces. A slight increase in the braking friction coefficient was observed for the measurement carried out in March. Perhaps the reason was a stronger solar operation, which dried the surface, especially grass.

The last of the parameters analyzed, the aligning moment, shows an interesting tendency (Figure 16). No significant impact of the weather period is observed during the months October–November, while a significant decrease in the *M_Z_* moment value in December was recorded. In March, the *M_Z_* value increased significantly compared to the December measurements.

## 4. Conclusions

An airfield experiment was conducted in order to measure landing gear wheel forces and moments. A five-element wheel force sensor was used to measure *F_X_* and *F_Z_* forces and *M_X_*, *M_Y_*, and *M_Z_* moments acting on an aircraft wheel during a low-speed taxiing. The measurement was conducted using the PZL 104 Wilga 35A as a test aircraft. The wheel force sensor was installed on the left landing gear in place of the wheel. Forces and moments were measured on a grassy surface of the Rzeszów Jasionka aerodrome, four times on days of autumn and spring months. The results show changes in wheel force and moment values during various maneuvers: Acceleration, turning left–right, and deceleration (braking). 

A significant influence of weather conditions on the values of determined coefficients *k_RR_* and *μ*, as well as on the aligning moment *M_Z_*, was noted. The rolling resistance coefficient increased during the considered weather period. However, the friction coefficient decreased, as did the stabilizing moment. The explanation for these trends is the increase in air and soil humidity, which translates into increased soil deformability and reduced adhesion.

Further research is planned in the area of applied machine learning to analyze the impact of unpaved airfield surface conditions on aircraft ground performance. First of all, we are planning to certify the wheel force system for flying in order to conduct flight tests. It is interesting how the forces and moments act during takeoff, as well as during landing, especially at touchdown. Also, we are planning to perform measurements on winter surfaces, different types of snow, ices, and at thawing conditions. Also, measuring wheel forces on higher grasses is also planned in the future. Another planned research is dynamic modeling of the airplane motion during takeoff and landing ground roll.

Regarding the wheel force sensor design, we are planning to develop and build a smaller version, possible to install into a wheel of a very light aircraft.

## Figures and Tables

**Figure 1 sensors-20-00227-f001:**
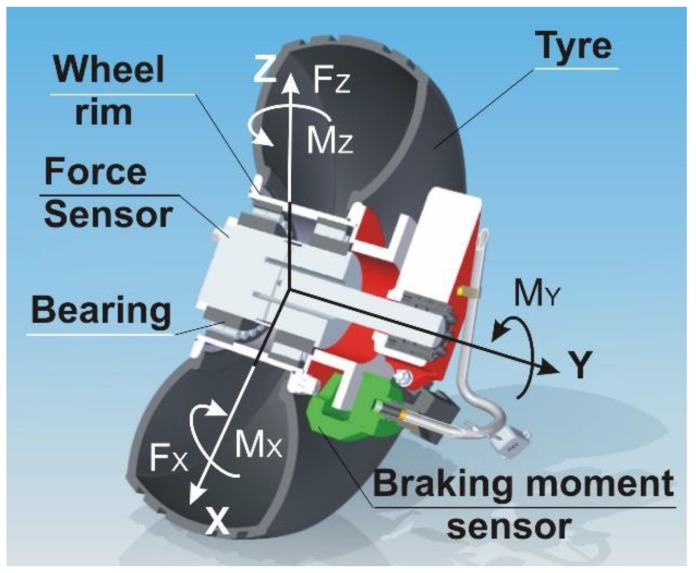
A schematic of the wheel sensor together with axes orientation and forces/moments system. The *X*, *Y*, and *Z* axes are mutually orthogonal.

**Figure 2 sensors-20-00227-f002:**
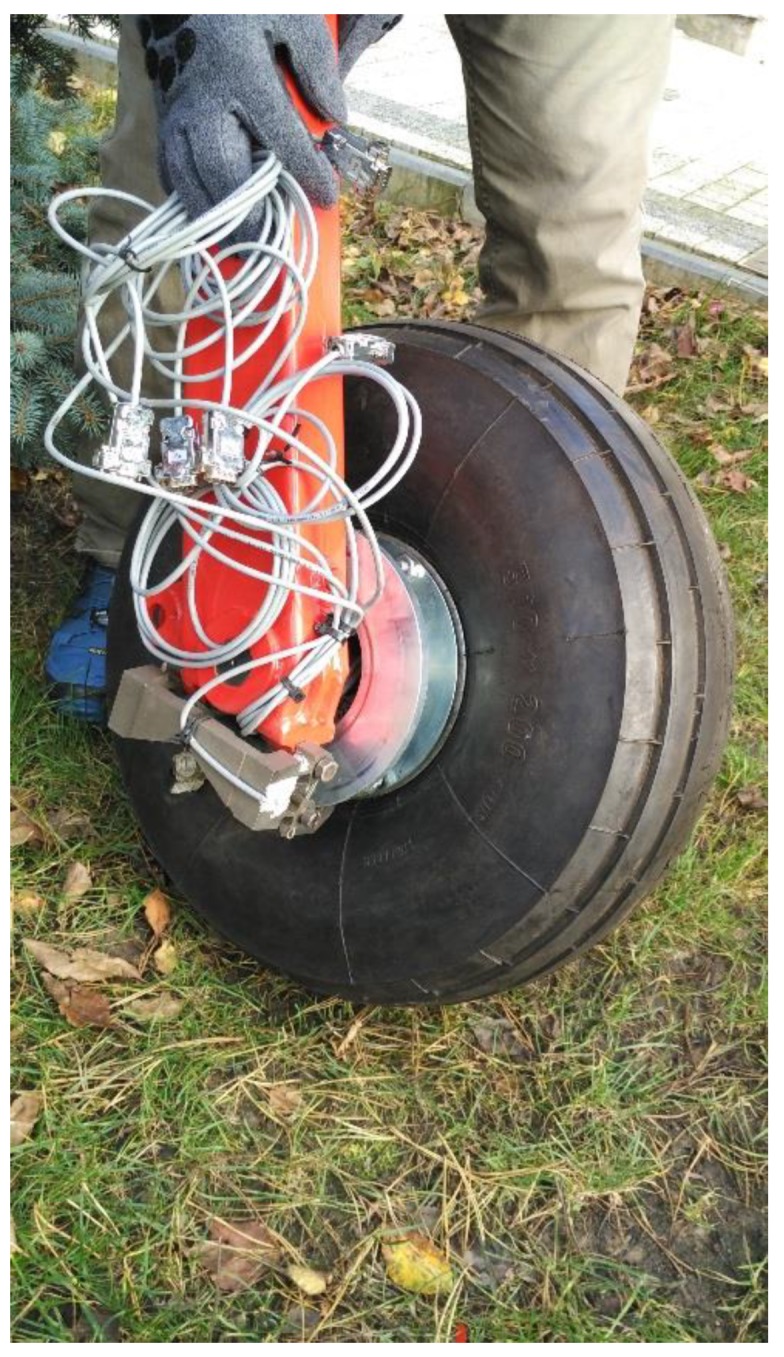
The complete wheel force sensor.

**Figure 3 sensors-20-00227-f003:**
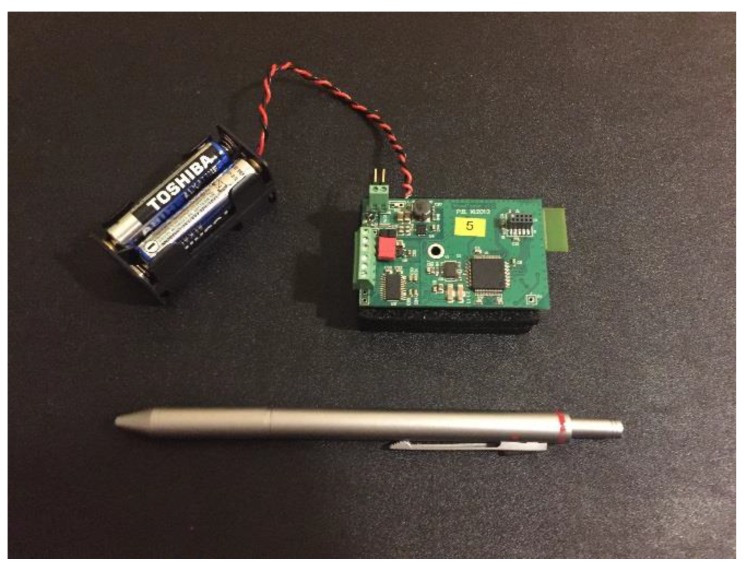
The heart of the microprocessor data transfer system, used in the wheel sensor system.

**Figure 4 sensors-20-00227-f004:**
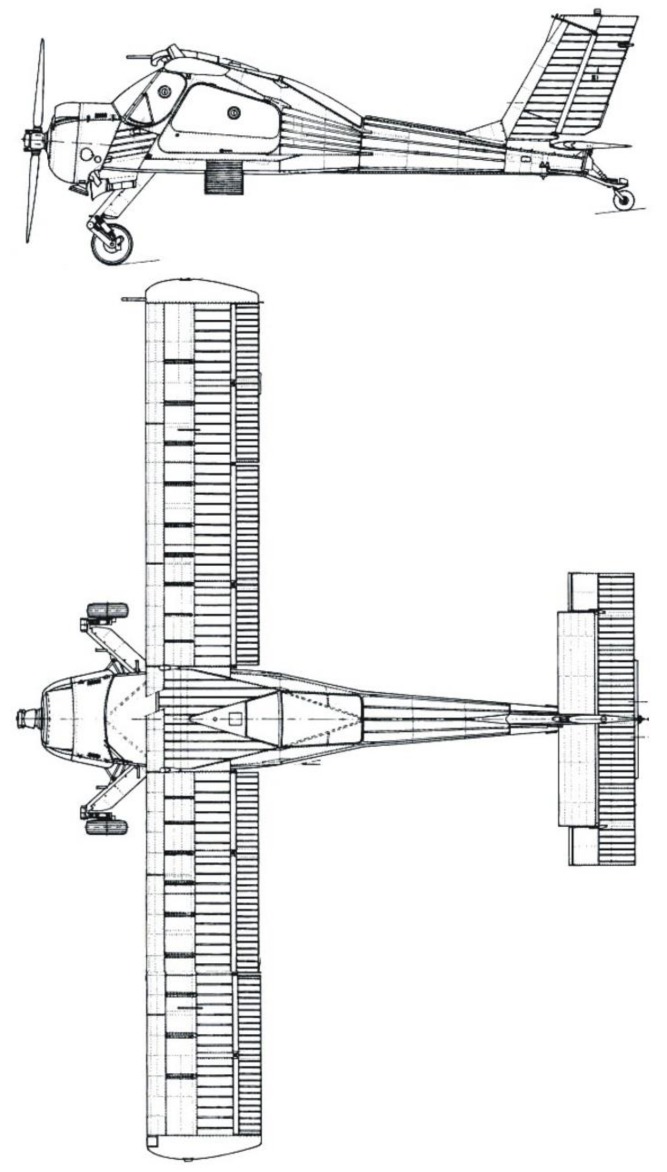
The PZL 104 Wilga 35A, used for the tests.

**Figure 5 sensors-20-00227-f005:**
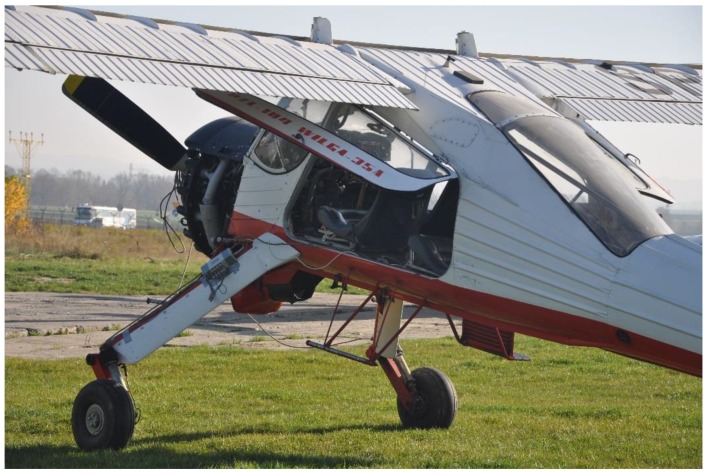
The wheel force sensor installed on the test airplane, ready for the tests.

**Figure 6 sensors-20-00227-f006:**
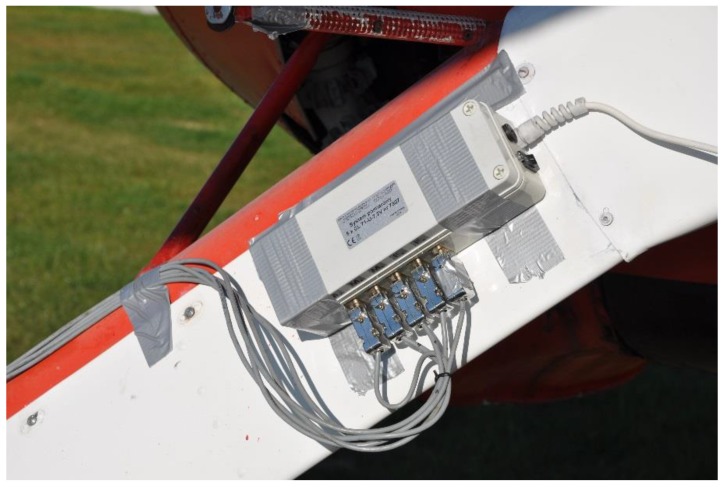
The electronics: Signal amplifiers and wireless data transfer system in the box, installed on the landing gear leg (a silver tape was used for quick installation).

**Figure 7 sensors-20-00227-f007:**
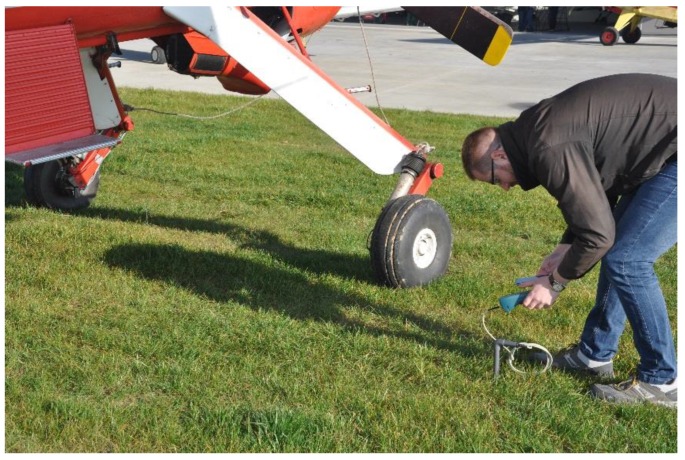
Measurement of soil moisture with the use of a handheld time-domain reflectometry (TDR) sensor.

**Figure 8 sensors-20-00227-f008:**
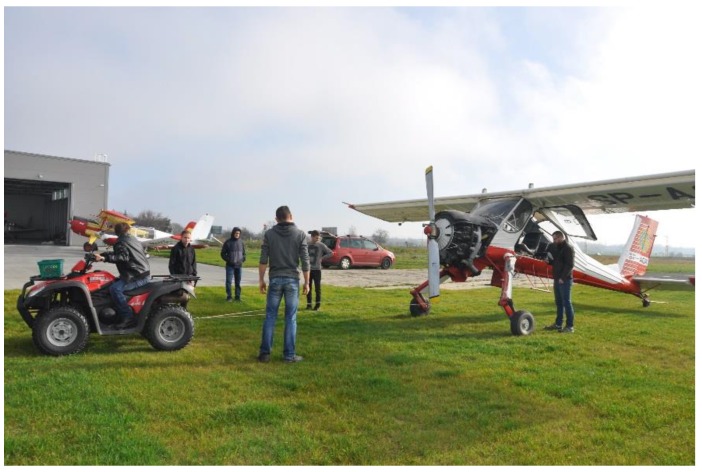
Starting a test run. The test airplane is pulled by an all-terrain vehicle (ATV). The test engineer is walking by the test airplane, collecting data wirelessly. Four persons onboard the airplane.

**Figure 9 sensors-20-00227-f009:**
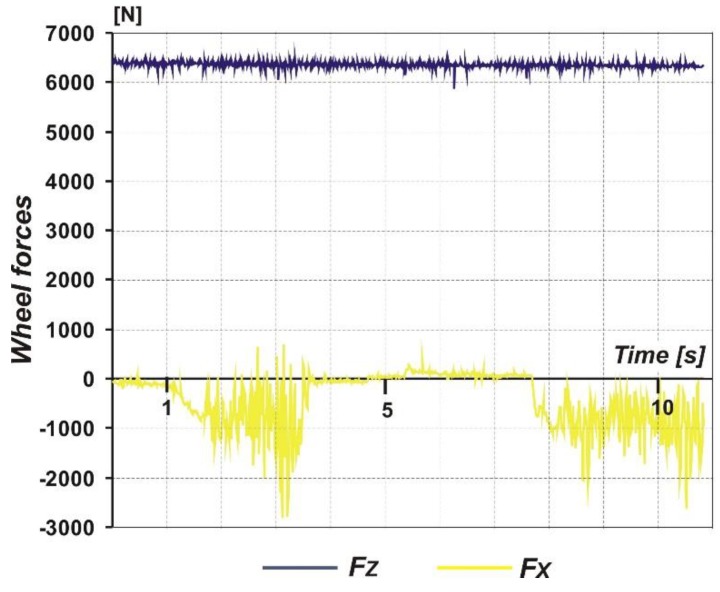
Sample result of the measurements: Time courses of wheel forces acting on the landing gear wheel.

**Figure 10 sensors-20-00227-f010:**
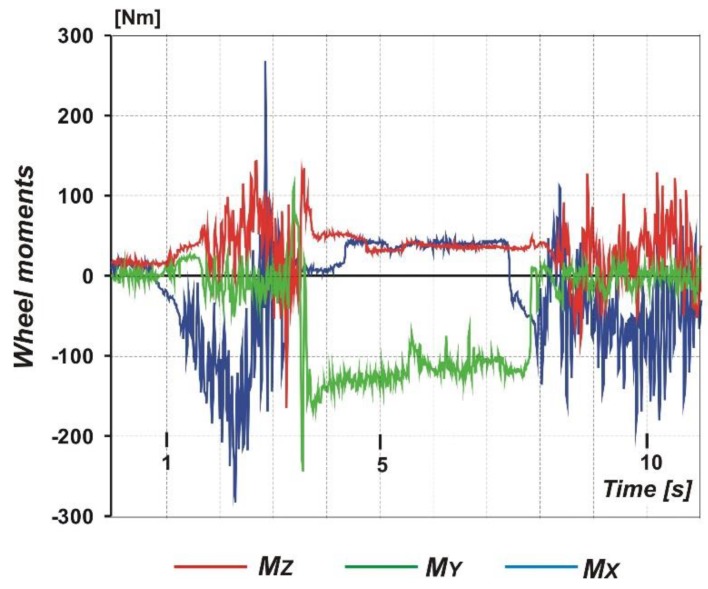
Sample result of the measurements: Time courses of three moments acting on the landing gear wheel.

**Figure 11 sensors-20-00227-f011:**
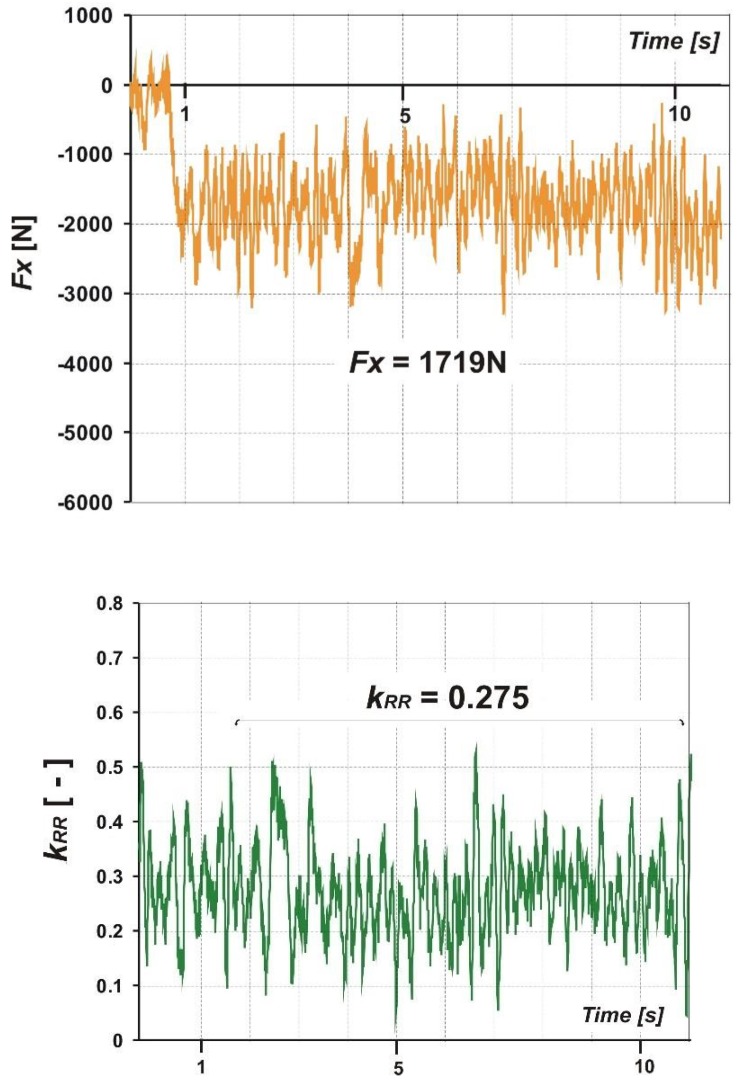
Sample courses of the measured horizontal force, *F_X_*, and calculated rolling resistance coefficient, *k_RR_*.

**Figure 12 sensors-20-00227-f012:**
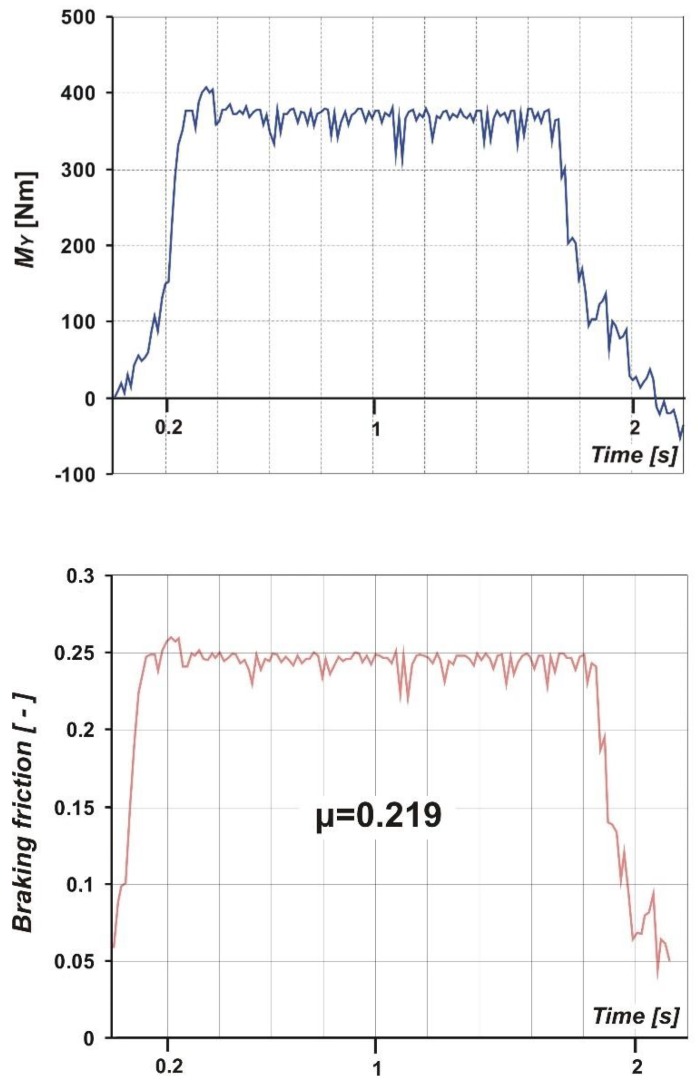
Sample courses of the measured braking moment, *M_Y_*, and braking friction coefficient, *μ*.

**Figure 13 sensors-20-00227-f013:**
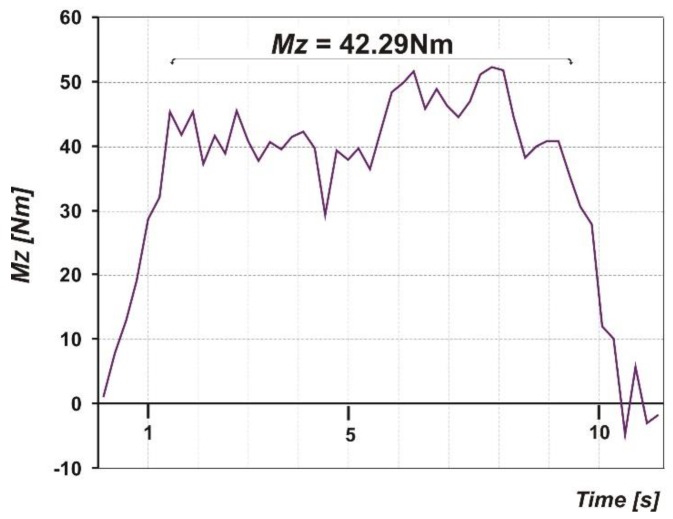
Sample course of the aligning moment, *M_Z_*.

**Figure 14 sensors-20-00227-f014:**
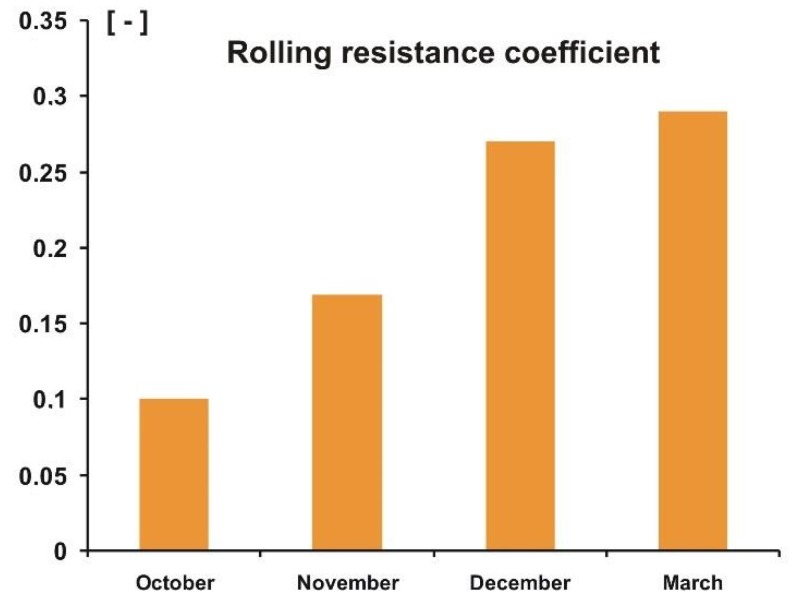
Average peak values of rolling resistance coefficient, *k_RR_*, for the four measurements.

**Figure 15 sensors-20-00227-f015:**
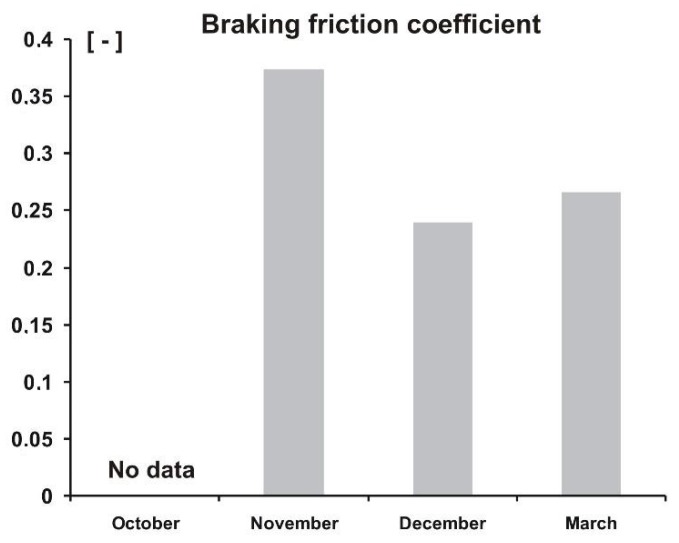
Average peak values of braking friction coefficient, *μ*, for the four measurements.

**Figure 16 sensors-20-00227-f016:**
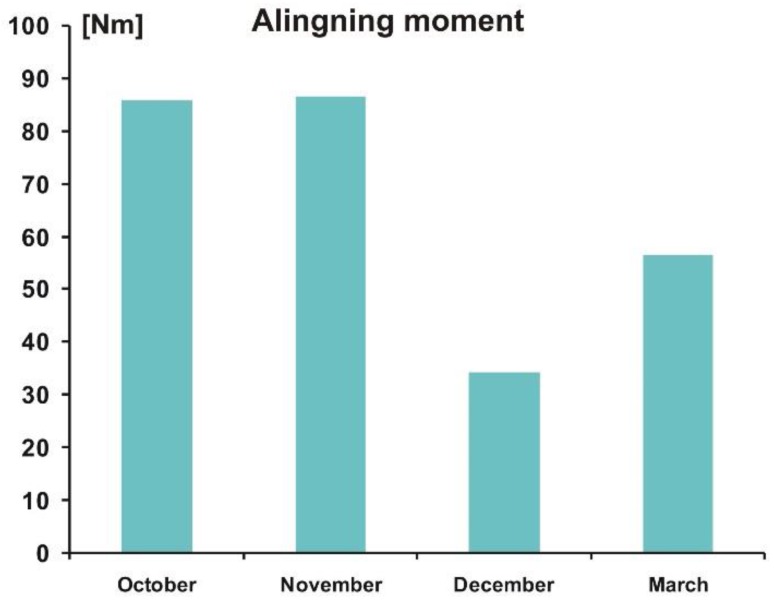
Average peak values of aligning moment, *M_Z_,* for the four measurements.

**Table 1 sensors-20-00227-t001:** Weather and soil conditions on the days of tests.

Weather	Air Temperature (°C)	Soil Moisture (%)	Soil Temperature (°C)
17 October 2018			
Sunshine, wind 5–7m/s S, SE	15	8.80	n.a.
5 November 2018			
Sunshine, wind 10-12 m/s W	12	18.80	18.3
5 December 2018			
Cloudy, 7/8 Cu, wind 2–3m/s W, NW	2,5	23.67	4.6
13 March 2019			
Cloudy, with break intervals	11	22.50	n.a.

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
