# Peer review of "Determining Wheel Forces and Moments on Aircraft Landing Gear with a Dynamometer Sensor"

_sensors, 2019, doi:10.3390/s20010227_

Round 1

Reviewer 1 Report

Dear authors

The topic discussed on the paper is of great interest to the readers. Likewise, sensor deployment in the final application is also interest. However, I noticed that 11 out of 15 references in your paper came from your own work; this is signaling that an inadequate review of literature has been done before writing this manuscript. Also, many of your self-cited articles discussed about the same topic, i.e. force and momentum measurements on grassy airfields. I understand that you have a defined research line, but every written article must add new knowledge, and not simply report new results with slight variations on measuring conditions. Other major changes are next listed:

Following with previous correction, what is the difference (and contribution) between this paper and your previous published papers? Specially, what is the difference between this manuscript and Refs. 8 and 13? Please add a clear statement in the introduction Section to reply this concern. Please include a figure that specifies the axes orientation (Fx, Fy, and Fz) in regard to sensor location. Why Fy measurements were not collected in Figure 8? Why Fx and Fz measurements are the only important ones? Note that is related with the previous concern. The quality of Figure 8 could be greatly enhanced, e.g. plane speed/acceleration could be simultaneously plotted with Fx and Fz measurements. Likewise, What is the measuring error resulting from Fx and Fz measurements? Have you considered the dynamic modeling of the plane? Such a model combined with acceleration measurements could provide a comparison for assessing the accuracy of your acquired data.

Minor corrections are next listed:

Line 51: depends must be followed by “on”. Please define what STOL is, line 86. Please provide location for the Rzeszow Jasionka aerodrome, line 114. Table 1: Please include a separate column for the date. Figure 8 is numbered twice in the paper. Is krr – as defined in Equation (1) – a concept from the authors or from someone else? Bear in mind that you statement “we define the rolling resistance coefficient…” suggests that Equation (1) was proposed by the authors, please clarify and provide proper citation if needed. A citation is needed for the statement in line 208.

Author Response

Dear Reviewer,

thank you very much for your time and effort to evaluate our manuscript. Your comments were very helpful.

With best regards,

Authors

Reviewer 2 Report

Dear Colegues,

the paper deals with airfield experiments conducted in order to measure landing gear wheel forces and moments and it is an extension of a paper already preented at the 6th IEEE International 15 Workshop on Metrology for Aerospace Metro Aerospace 2019, Totino, Italy in June 2019.

The paper does not show great innovations because it shows test realized in field for the measurement of gear wheel forces giving a great importance to the experimental set up and test more than a theoretical developing. Being me an experimental scientist, I apprecciated so much all the efforts made by the research group to realize the test in field that involve the use of an airplane and the use of a instrumentation good for airplanes. In my opinion, the paper can be useful for other researchers in the world that work on the same argument giving suggestions on how a so dynamic and articulate test can be set up.

Therefore, even if the paper does not show theoretical innovations, it represents a challenge in practical measurements and, in this sense could be considered innovatiove. Moreover the data picked up and shown in the results are not much present in literature nad tehrefore very interesting.

Despite my good impression an my will to go versus an acceptation of the paper, it needs some important improvements:

1) (Mandatory) Sensors is an important Measurement Journal and the writing of the measurement unit must follow the international standard; so, please depart the numbers from the measurement unit, e.g., if you want to write 500×200 millimeters, it must be written 500X200 mm and non 500X200mm, so as 1150kg must be changed in 1150 kg. Please check all and correct.

1 (bis) (Mandatory) Please avoid stupid errors as 11,12 meters, that mus be written 11.12 m and non 11,12m. Please check all and correct.

1 (tris) (Mandatory) Please avoid to use measurement units not present in the International Measurement System as "HP" that must be replaced with kW. Please check all and correct

1 (tetris) (Mandatory) Please avoid to use symbols not present in the International Measurement System: in this sense watch the symbol used for the degrees at row 89, please write in this way: 18º. Check all and correct.

2) Formatting better table 1

3) Formatting better the functions, e.g. for formula (1), the number of the formula must be write at the roght end of the row. Check all and correct.

4) (Strongly suggested) As said, Sensors is an important scientific Journal in which a rich bibliography that increase the overall value of the paper, is always welcome. Even if the paper is an extension of a Congress one, an effort to provide other paper can be done. Here I suggest three papers that you are completely free to consider or not, but, anyway, please increase the number citations:

a)  in the first two papers the field test assume a great role in the paper:

Hinüber, E.L.V., Reimer, C., Schneider, T., Stock, M.
INS/GNSS integration for aerobatic flight applications and aircraft motion surveying
(2017) Sensors (Switzerland), 17 (5), art. no. 941, . Cited 18 times.
https://www.scopus.com/inward/record.uri?eid=2-s2.0-85018305039&doi=10.3390%2fs17050941&partnerID=40&md5=5b4690558f23032a4d73c22f28073e9e
DOI: 10.3390/s17050941

b) Baiocchi, V., Napoleoni, Q., Tesei, M., Costantino, D., Andria, G., Adamo, F.
First tests of the altimetric and thermal accuracy of an UAV landfill survey
(2018) 5th IEEE International Workshop on Metrology for AeroSpace, MetroAeroSpace 2018 - Proceedings, art. no. 8453601, pp. 403-406. Cited 2 times.
https://www.scopus.com/inward/record.uri?eid=2-s2.0-85053882833&doi=10.1109%2fMetroAeroSpace.2018.8453601&partnerID=40&md5=583bacd944ce4a0df45b361e7d19c1a8
DOI: 10.1109/MetroAeroSpace.2018.8453601

c) the third paper shows an application of ho to check and control the anti blocking system for UAV

Petritoli, E.R., Leccese, F., Cagnetti, M.
Takagi-Sugeno Discrete Fuzzy Modeling: An IoT Controlled ABS for UAV
(2019) 2019 IEEE International Workshop on Metrology for Industry 4.0 and IoT, MetroInd 4.0 and IoT 2019 - Proceedings, art. no. 8792915, pp. 191-195.
https://www.scopus.com/inward/record.uri?eid=2-s2.0-85071535006&doi=10.1109%2fMETROI4.2019.8792915&partnerID=40&md5=25a148359fd2b6280bf982dc3c217047
DOI: 10.1109/METROI4.2019.8792915

I give major revision only because the Journal does not offer an intermediate evaluation level between major and minor.

Author Response

Dear Reviewer,

thank you very much for your valuable comments on our manuscript. We hope our rely will satisfy you.

Warmest regards,

Authors

Round 2

Reviewer 1 Report

Dear authors

Suggested changes have been donde. New references have been added (as suggested in first review).

Author Response

Dear Reviewer,

thank you very much again for your time and effort to review our revised manuscript.

We're happy that you're satisfied with our modificaitons done to the original text. Just for your information, we've further modified the Introduction section, as a reply to Reviewer #2 suggestions (we've added some more references).

Thank you very much.

We wish you a Merry Christmas and a Happy New Year.

Warmest regards,

authors

Reviewer 2 Report

I confirm my good opinion regards the experimental approach followed by the Authors.

The improvements implemented in this new version surely give more strenght to the paper and make it clearer.

In order to improve the final version, although the bibliography is surely improved, I think that a little effort could be done by the Collegues.

Here, I suggest other two papers:

1) Pecora, A., Maiolo, L., Minotti, A., De Francesco, R., De Francesco, E., Leccese, F., Cagnetti, M., Ferrone, A.  Strain gauge sensors based on thermoplastic nanocomposite for monitoring inflatable structures (2014) 2014 IEEE International Workshop on Metrology for Aerospace, MetroAeroSpace 2014 - Proceedings, art. no. 6865899, pp. 84-88.  DOI: 10.1109/MetroAeroSpace.2014.6865899   that is an example of use of innovative strain gauge in aerospace and the second that is an innovative georeferencing system for the guidance, navigation control of UAV   2) Eling, C., Klingbeil, L., Kuhlmann, H.
55228841300;23389643100;55807201200;
Real-time single-frequency GPS/MEMS-IMU attitude determination of lightweight UAVs
(2015) Sensors (Switzerland), 15 (10), pp. 26212-26235.390/s151026212
  In this sense, I'll give a minor revision

Author Response

Dear Reviewer,

thank you very much for your valuable comments. We've included two references based on your suggestions and three references more, that we've found interesting and improving the overall quality of the manuscript.

We hope those modifications will satisfy you. Changes made in the review round 2 have been highlighted in navy blue colour - please see the revised version of  the text.

Thank you once again.

We wish you a Merry Christmas and a Happy New Year.

Best regards,

authors